# Development of the Physics–Based Morphology Model as the Platform for the Optimal Design of Beach Nourishment Project: A Numerical Study

Yong Jun Cho 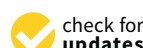

Department of Civil Engineering, University of Seoul, Seoul 02504, Korea; young@uos.ac.kr

**Abstract:** In this study, a physics-based morphology model is developed and to test the feasibility of the morphology model proposed in this study as the platform for the optimal design of the beach nourishment project, the beach restoration process by the infra-gravity waves underlying the swells in a mild sea is numerically simulated. As a hydrodynamic module, the IHFOAM wave toolbox having its roots in the OpenFoam is used. Speaking of the morphology model, a transport equation for suspended load and the Exner type equation constitute the morphology model. In doing so, the probability theory first introduced by Einstein and the physical model test by Bagnold are used as the constituent sub-model of the morphology model. Numerical results show that among many flow features that are indispensable in forming sand bars over the flat bottom and swash zone, the partially skewed and asymmetric bottom shearing stresses, a shoreward Stokes drift near the free surface, boundary layer streaming near the seabed, and undertow toward the offshore were successfully simulated using the morphology model proposed in this study. It was also shown that plunging type breaker occurring at the final stage of the shoaling process, and its accompanying second breaker, sediment entrainment at the seabed, and the redistribution of suspended load by the down rush of preceding waves were successfully reproduced in the numerical simulation, and agreements with our experience in the field were very encouraging. In particular, the sand bar formation process over the flat bottom and backshore were successfully reproduced in the numerical simulation, which has been regarded as a challenging task.

**Keywords:** beach restoration; physics-based morphology model; IHFOAM; RANS; Einstein's probability theory; Bagnold's physical model test

## 1. Introduction

Over the last few years, beaches along the east coast of South Korea have been suffering from severe erosion since the quasi-equilibrium water environment that they once enjoyed has been damaged by poorly executed development [1,2]. There is a growing consensus that the conventional countermeasures against beach erosion, such as a detached breakwater, LCB (low crested breakwater), groin, and jetty, are inadequate for restoring the damage to the quasi-equilibrium water environment and would only result in additional scouring or siltation at nearby beaches [1]. Therefore, soft structures such as beach nourishment are emerging as a promising alternative to conventional beach stabilization methods. Even though a quasi-equilibrium water environment cannot be restored by beach nourishment, a beach's natural characteristics can be sustained through periodic nourishment [1,2].

Despite the significant progress on wave drivers over the last decades, prediction of the erosion rates of a nourished beach exposed to storm waves remains a formidable challenge. However, the suspension taking place over a swash zone supplies most of the sediments available along the shore, and as a result, an accurate estimation of the beach's erosion rate during storm waves is a crucial

design factor for the optimal design of beach nourishment project [3]. These difficulties stem from our limited understanding of boundary layer flow and its associated bottom shearing stress over the surf and swash zone [4]. This poor understanding can partially be attributed to the habitual referencing of the depth- and time-averaged approach in past studies. Most physics-based morphology models in the current literature heavily rely on the depth-averaged approach based on the assumption of negligible vertical acceleration or hydrostatic pressure distribution, as in the Boussinesq-type equation and NSW (nonlinear shallow water equation) owing to the restrictions of computational resources [5]. Moreover, it is often assumed in the depth-averaged approach that the horizontal flow velocity distribution along the vertical direction follows $(z + h)^2$. However, these assumptions are hardly fulfilled in a rapidly accelerating and slowly decelerating flow over the surf and swash zone. As a result, with these morphology models, the offshore directed return flow near the seabed cannot be accounted for, and the entrainment of the sediment particles over the swash zone cannot be described accurately enough to implement the beach nourishment efficiently. These limitations have led us to a poor understanding of flow over a swash zone and highlighted the need to relax the assumptions mentioned above [6,7].

Lately, to overcome these intrinsic limitations of the depth-averaged approach, Dalrymple and his colleagues at Johns Hopkins University directly tackled the Navier–Stokes equation coupled with the transport equation for the suspended load with the constant Smagorinsky model as a turbulence closure [8]. In this study, a plunging breaker, its accompanying energetic suspension of sediment from the seabed, and the redistribution of suspended sediments by a down-rush of preceding waves were successfully simulated. However, Zou et al. [8] used the pickup function by Van Rijn [9] to evaluate the sediment entrained from the bottom over the swash zone whenever an effective Shield's parameter surpassed a threshold and redistributed the entrained sediment according to the suspended load transport equation. However, it should be noted that Van Rijn's [9] pickup function is for steady, uniform flow over a constant depth, and its application to the swash zone should be made with caution and subject to further tests. Furthermore, the instantaneous bottom sediment concentration was estimated by assuming a local balance between the deposition and pickup rate.

In the light of the facts mentioned above, it can be easily perceived that with the morphology model based on the Van Rijn's [9] pickup function and local balance, the erosion rates of a nourished beach exposed to storm waves cannot be estimated accurately enough to ensure the optimal design of beach nourishment project. Furthermore, Zou et al. [8] assumed turbulent intensity as being uniform throughout the surf zone as manifested in the constant Smagorinsky model. Therefore, their model cannot describe locally generated turbulence, its ensuing dispersion, and dissipation, as is the case with a plunging-type breaker. With the advent of the OpenFoam, there are many attempts to extend our understanding of the flow over the swash zone by directly solving the RANS (the Reynolds averaged Navier–Stokes equation) [10–12]. Among these, Jacobsen and Fredsoe's [10] model is the most comprehensive morphology model to describe the formation of an offshore bar and its seasonal migration. Jacobsen and Fredsoe's [10] model will be hailed as the beginning of the real 3D physics-based morphology model, but it is not free of flaws. Of these flaws, excessive filtering enforced at each time step in order to overcome the numerical instability problem of the bed morphology model noticeably stands out, but this filtering has no solid background that can justify its use and should be avoided as possible. In this rationale, Cho [13,14] proposed a revised physics-based 3D morphology model to be free from the ambiguity that occurred by artificial damping based on the filtering and proceeded to carry out the numerical simulation to demonstrate the effectiveness of the revised morphology model. In this study, Cho [14] successfully demonstrated that sediments suspended at the foreshore by wave breaking are gradually drifted toward a shore and accumulated in the process of up-rush, which eventually leads to the formation of a swash bar. It was also shown that, with the revised morphology model, the formation process of sand waves or ripples on the flat bottom was reproduced in the numerical simulation, which has been regarded as a very challenging task in the community of coastal engineering. However, how or under what wave conditions sand waves or ripples are formed on the flat bottom has not been sufficiently addressed. Hence, it is unclear

whether sand waves or ripples observed in the numerical simulation were by chance or had a robust physical basis.

As a result, it is of great engineering value from the perspective of the development of a physics-based morphology model robust enough to work as the platform for the optimal design of beach nourishment project to discuss what distinguishes Cho [14] from the other morphology models of, e.g., Zou et al. [8] and Jacobsen and Fredsoe [10]. Cho's [14] morphology model consists of RANS coupled with a transport equation for suspended load, and Exner type equation derived from the principle of the sediment budget with the bed load being accounted for. Recalling that these equations mentioned above were well known relatively early in the coastal engineering community, there is nothing new. However, it is how to estimate the bottom shearing stress concerned with the bedload transport rate and bottom sediment concentration needed to close the boundary value problem for the suspended load that differentiates Cho [14] from the other studies. Bottom sediment concentration was estimated using the instantaneous bottom shearing stress directly evaluated from the numerically simulated velocity profile rather than the conventional quadratic law using the period averaged velocity and frictional coefficient as in the Van Rijn's [9] pickup function. Cho [14] classified the instantaneous bottom shearing stress as three types. The first one is the shearing stress acting on the immobile sediment layer, and the second one is the shearing stress acting on the sediment particles moving as bedload. The shearing stress involved in saltating sediment particles from the seabed constitutes the remaining. Of these shearing stresses, the last two shearing stresses are modeled, respectively, following the probability theory first introduced by Einstein [15] and physical model test by Bagnold [16]. Even though it has been a while since the robustness of these models mentioned above was known in the coastal engineering community, these models have never been fully integrated into a numerical simulation based on the phase-resolving wave driver like RANS coupled with physics-based morphology model due to restrictions of computational resources.

In the light of the discussions mentioned above, it can be easily perceived that numerical duplication of sand waves or ripples formed on the flat bottom, which is regarded as the benchmark test of a physics-based 3D morphology model to work as the platform for the optimal design of a beach nourishment project, would be plausible utilizing the RANS and physics-based morphology model having the probability theory first introduced by Einstein [15] and the physical model test by Bagnold [16] as the constituent sub-model.

In this rationale, this study intends to test the hypothesis mentioned above, using the numerically simulated beach profile exposed to waves using the revised physics-based morphology model by Cho [13,14] that consists of the phase-resolving RANS coupled with a transport equation for suspended load, and morphology equation of Exner type.

This article is structured as follows. For the sake of self-containment, Section 2 provides details about how the morphology model consisting of the Van Rijn's [9] pickup function preferred in past studies evolved from the perspective of the quadratic frictional law based on the period average velocity and frictional coefficient, and in doing so, some efforts are made to review the limitations of this methodology critically. In Section 3, the numerical model is described, and the performance of the morphology model proposed in this study is shown in Section 4. Section 4 also contains the discussion of the numerical results. In Section 5, the conclusion and recommendations are presented.

## 2. Sediment Transport Model

In most physics-based morphology models available in the current literature, the incipient sediment motion is determined by the Shields parameter, which is the ratio of the shearing force acting on the seabed to the submerged weight of sediment particle, and the incipient sediment motion commences whenever the Shields parameter exceeds a threshold. The Shields parameter $\theta'$ can be written as:

$$\theta' = \frac{\tau_b}{\rho(s-1)gd} \tag{1}$$

where $\tau_b$ is the bottom shearing stress, $s$ is the specific gravity, $\rho$ is the seawater density, and $g$ and $d$ are the gravitational acceleration and the sediment diameter, respectively.

## 2.1. Transport Eq. of Suspended Load

The concentration of suspended load within the water column can be described by the transport equation, which can be written as:

$$\frac{\partial c}{\partial t} + \nabla \cdot \left[\left(\gamma u + w_f\right)c\right] = \nabla \cdot [\gamma(\nu + \nu_t)\nabla c] \tag{2}$$

where $c$ is the concentration of suspended load per unit volume, $w_f$ and $u$ are the falling velocity of sediment particles and flow velocity vector, respectively. $\gamma$ is the VOF coefficient and has a unit value within the fluid and converges to a value of zero when not in the fluid. In this study, the location of free surface is defined as $\gamma = 0.5$. In the Equation (2), $\nu$ and $\nu_{tub}$ are kinematic viscosity and eddy viscosity, respectively. To close the boundary value problem concerned with the suspended load in Equation (2), the bottom sediment concentration $c_b$ is needed and has been estimated by assuming a local balance between the deposition rate and the pickup rate.

## 2.2. Review of Previous Studies on $c_b$ Based on the Pickup Function by Van Rijn (1984)

The bottom sediment concentration required for the analysis of Equation (2) has been assumed to depend on the Shields parameter. As Engelund and Hansen [17] pointed out, the bottom shearing stress can be classified as the form drag and skin friction. Among these, the form drag is generated owing to the difference in pressures between the up-wave and down-wave bed forms and does not affect the stability of individual surface sediment particles.

As a result, the Shields parameter is modified as follows:

$$\theta' = \frac{\tau'}{\rho g(s-1)d} \tag{3}$$

where $\theta'$ is known as skin friction Shields parameter, and skin friction $\tau'$ is referred to as the effective stress concerned with sediment transport. However, there is some indication that the skin friction even on flat beds under waves would not be completely effective in transporting sediment owing to the momentum transferred to the immobile sediment layer via the colliding between the sediment particles. Unfortunately, this momentum transfer is very difficult to estimate, and related data is very scarce. Hence, a generally accepted method for calculating the skin friction on a bed of highly mobile sand under waves is not yet available, which prompted Madsen and Grant [18] to suggest that sediment mobility be estimated in terms of grain roughness Shields parameter. This approach has since been quite popular. The value of $2.5d_{50}$ is usually adopted for the grain roughness of flat-bed with a median diameter of $d_{50}$ [17,19], and the grain roughness Shields parameter $\theta_{2.5}$ is defined as follows:

$$\theta_{2.5} = \frac{\frac{1}{2}f_{2.5}(A_b\omega)^2}{(s-1)d} \tag{4}$$

where $f_{2.5}$ is the grain roughness friction factor given by the Swart's formula with $r = 2.5d_{50}$ that can be written as:

$$f = \exp\left[5.213\left(\frac{r}{A_b}\right)^{0.194} - 5.977\right] \tag{5}$$

In Equations (4) and (5), $A_b$ is the semi-excursion amplitude of water particle near the seabed, $\omega$ is the wave frequency, $r/A_b$ is the relative roughness, and $r$ is the Nikuradse roughness height.

The instantaneous $c_b(t)$ as waves proceed can be determined by assuming local balance between the deposition rate and the pickup rate as follows:

$$c_b = \frac{p(t)}{w_o} \tag{6}$$

where $p(t)$ is the pickup function given by van Rijn [9] and $w_o$ is the settling velocity. The pickup function given by Van Rijn [9] for unsteady flows can be written as follows:

$$p(t) = 0.00033\left(\frac{\theta'-\theta_c}{\theta_c}\right)^{1.5}\frac{(s-1)^{0.6}g^{0.6}d^{0.8}}{\nu^{0.2}} \quad For \ \theta > \theta_c$$

$$p(t) = 0 \qquad\qquad\qquad\qquad\qquad For \ \theta < \theta_c \tag{7}$$

In Equation (7), $\theta'(t)$ and $\theta_c$ denote the effective Shields parameter and the critical Shields parameter, respectively. Although $p(t)$ is based on van Rijn's experiments on scour rates under steady flow, Zou et al. (2005) extended its application to the surf zone with a slight modification, as in Equation (7), with the grain roughness Shields parameter $\theta_{2.5}$ chosen as the effective Shields parameter. However, in their study, Zou et al. [8] did not give any information about the thickness of the boundary layer, which is crucial for the estimation of the semi excursion amplitude $A_b$ and the effective Shields parameter. Hence, there was huge room for controversy considering that the boundary layer over the swash zone is not easy to identify.

### 2.3. New Suggestion for the Estimation of Shields Parameter

Flow over the swash zone accelerates rapidly and decelerates slowly in a saw-tooth wave profile with a steep front. For these short durations, the boundary layer has less time to grow during the rapid shoreward acceleration; hence, the shoreward bed shear stress is considerably higher than its offshore directed counterpart, as reported by King [20]. The flow complexity reaches its extreme in the swash zone due to the offshore directed return flow near the seabed, shoreward drift near the surface, and plunging breaker [21], which is beyond the present state of the art. Very little is known about the boundary layer flow and its associated bottom shear stresses over the swash zone to even attempt a description of the underlying mechanism of sediment transport.

Our understanding of flow over the swash zone has been slowed down by our habitual referencing to the depth-averaged approach accompanied by the conventional quadratic frictional law and friction coefficient, which has been very popular over the last decades. On the other hand, the bottom shearing stress $\tau_b$ can be directly estimated as follows, if a numerically simulated 3-D flow field is available, that is not a case in the depth-averaged approach:

$$\boldsymbol{\tau}_b = \rho(\nu + \nu_{turb})\left(\frac{\partial u}{\partial z} + \frac{\partial w}{\partial x}\right) \tag{8}$$

From Equation (8), the effective Shields parameter $\theta'$ can be determined as follows:

$$\theta' = \frac{\tau_b}{\rho g(s-1)d} \tag{9}$$

By doing so, the controversial issue concerned with the ambiguity in the estimation of the grain roughness friction factor $f_{2.5}$ in Equation (4) can be addressed.

### 2.4. $c_b$ Based on the Probability Theory Introduced by Einstein and the Physical Model Test by Bagnold

It was Einstein [15] who first introduced probability theory in the computation of $c_b$ and succeeded in obtaining physically meaningful results. Later, Bagnold [16] points out that, even if the Shields parameter exceeds the critical value, not all of the sediment layer at the sea bottom is peeled off,

some of the shearing-stresses acting on the sea bottom are transferred to the immobile sediment layer via the colliding between the sediment particles. In this study, Bagnold [16] has verified the hypothesis mentioned above through a series of sophisticated experiments (see Figure 1).

The shear stress $\tau_G$ and the normal stress $\sigma_G$ transmitted to the immobile bottom sediment layer observed by Bagnold [16] during the experiment can be described by the dimensionless numbers that can be written as follows:

$$N = \frac{\sqrt{\lambda} s d^2}{v} \frac{du}{dz} \tag{10}$$

$$G = \frac{d}{v} \sqrt{\frac{\sigma}{\lambda} \frac{s}{\rho}} \tag{11}$$

Here, $\lambda$ is the linear sediment concentration at the bottom introduced by Bagnold [16] and denotes the separation distance between the sediment particles that is inversely proportional to sediment concentration.

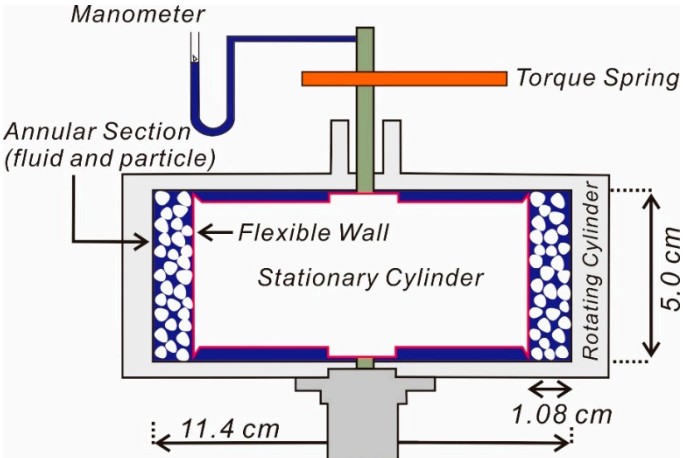

**Figure 1.** Experimental apparatus used by Bagnold [16] to measure the shear and normal forces in a sheared suspension.

If n sediment particles are migrating as bed load, the instantaneous shearing stress $\tau'$ acting on the sea bottom can be written as (see Figure 2):

$$\tau' = \tau_G + nF_D \tag{12}$$

where $\tau_G$ is the shearing stress acting on the immobile sediment layer and $F_D$ is the flow-induced drag force acting on the sediment particle moving as the bed load, which can be written as:

$$F_D = \frac{1}{2} C_D \rho d U^2 \tag{13}$$

The drag force $F_D$ can be approximated in terms of the bottom friction force proportional to the weight of sediment particles such as:

$$F_D \simeq \rho g (s-1) \frac{\pi}{6} d^3 \mu_d \tag{14}$$

In Equations (13) and (14), $C_D$ is the fluid drag coefficient, $\mu_d$ is the kinetic friction coefficient and has a value of 0.6.

Upon noting that the shear stress $\tau_G$ acting on the immobile sediment layer cannot exceed the critical shear stress, $\tau_G$ can be approximated by a critical shear stress $\tau_C$. In this case, Equation (12) can be rewritten as:

$$\tau' = \tau_C + \rho g(s-1)\frac{\pi}{6}nd^3\mu_d \tag{15}$$

After normalizing both sides of the Equation (15) by $\rho g(s-1)d$, the following relationship can be obtained:

$$\theta' = \theta_c + \frac{\pi}{6}nd^2\mu_d \tag{16}$$

Upon recognizing that the number of sediments per unit area is $1/d^2$, and $nd^2$ corresponds to the probability $pEF$ that the bottom sediment is moving as bed load, the following relationship can be obtained:

$$\theta' = \theta_c + \frac{\pi}{6}\mu_d pEF \tag{17}$$

After some elaboration to correct the over shooting problem of $pEF$ when the shearing stress is relatively large, the probability $pEF$ of the bottom sediment grains to be transported along the seabed can be defined as follows:

$$pEF = \left[1 + \left(\frac{\frac{1}{6}\pi\mu_d}{\theta' - \theta\prime_c}\right)^4\right]^{-1/4} \tag{18}$$

When hydrodynamic force increases as flow accelerate, some sediment particles start to move in the form of a suspended load as well. In this case, the instantaneous bottom shearing stress is modified as follows:

$$\tau' = \tau_c + nF_D + \tau_F \tag{19}$$

where $\tau_F$ denotes the shearing stress involved in the saltating the sediment particles off the bottom, and following the Bagnold [16] model, $\tau_F$ can be written as:

$$\tau_F = 0.013\rho s(\lambda d)^2\left(\frac{du}{dz}\right)^2 \tag{20}$$

Upon assuming that the flow velocity at the boundary layer near the sea bottom follows the logarithmic distribution, the velocity gradient $du/dz$ in Equation (20) can be written as:

$$\frac{du}{dz} \simeq \frac{U'_f}{\kappa z} \tag{21}$$

where $\kappa$ is the universal constant of von Karman (0.4). Since the thickness of layer where the flow velocity follows the logarithmic distribution is limited, the distance from the bottom $z$ in Equation (21) can be approximated as:

$$z = \alpha_1 d \tag{22}$$

where $\alpha_1$ has a value of $O(1)$.

After substituting Equations (20)–(22) into Equation (19) and some elaboration, the following relationship can be obtained:

$$\theta' = \theta_c + \frac{\pi}{6}\mu_d pEF + \frac{0.013}{\kappa^2\alpha_1^2}s\theta'\lambda_b^2 \tag{23}$$

From Equation (23), the instantaneous linear sediment concentration at the bottom $\lambda_b$ can be written as

$$\lambda_b^2 = \frac{\kappa^2\alpha_1^2}{0.013s\theta'}\left(\theta\prime - \theta_c - \frac{\pi}{6}\mu_d pEF\right) \tag{24}$$

From Equation (24), the instantaneous volumetric bottom concentration $c_b$ used in this study is written as [22]:

$$c_b = \frac{c_0}{\left(1 + \frac{1}{\lambda_b}\right)^3}$$

(25)

where $c_0$ is the maximum suspended load concentration, which is known to have a value of 0.3 when most densely packed.

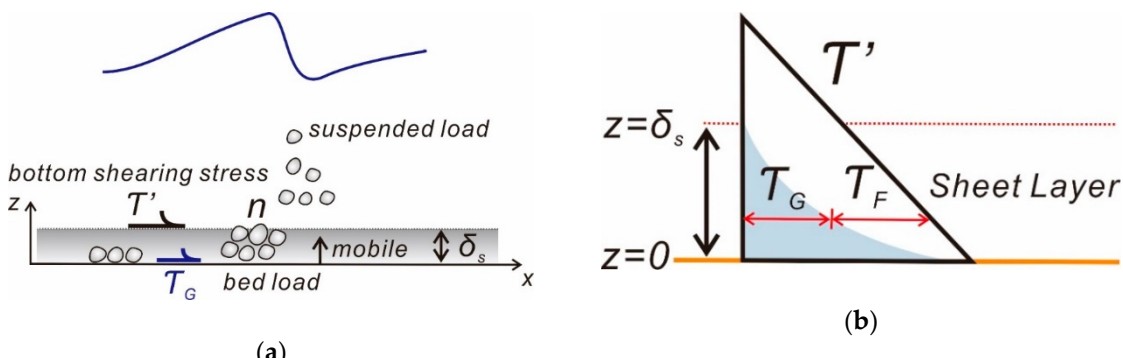

**Figure 2.** Partition of bottom shearing stresses into the ones carried by fluid and soil granular particle, and a distribution of shearing stress. (**a**) Partition of bottom shearing stresses into the ones carried by fluid and soil granular particle, and definition sketch of thickness of sheet layer. (**b**) distribution of shearing stress carried by fluid and soil granular particle within the sheet layer.

### 2.5. Bed Load Transport Rate Based on the Probability Theory Introduced by Einstein and the Physical Model Test by Bagnold

With the instantaneous bottom shearing stress $\tau_b$ being available from the numerically simulated flow field using the phase-resolving wave driver like RANS, the bed load transport rate $q_b$ can be estimated such as:

$$q_b = \frac{1}{6}\pi d p E F u_b$$

(26)

where $d$ is the diameter of the sediment, $pEF$ is the probability of the sediment transfer defined in Equation (18), and $u_b$ is the transport velocity of bed load.

### 2.6. Bed Load Transport Velocity

The most formidable challenge in the analysis of bedload lies in evaluating the bed grain velocity and the flow velocity in the sheet layer. The bed load moves at a small fraction of the fluid flow velocity randomly fluctuating by strong rhythmic vortices emitted at the wake of a bedform like a ripple that is the coherent structure of the seabed. To overcome these difficulties, the drag force acting on the sediment has been estimated using the time average flow velocity rather than the instantaneous flow velocity. A similar approach can be found in the studies on the bedload transport rate and representative flow velocity near the bottom, usually taken as multiple of the frictional velocity $(\alpha U_f')$ [4]. On the contrary, in the numerical simulation based on the phase-resolving wave driver like the RANS, the information concerned with flow velocity near the bottom boundary layer is available, and as a result, the bed load transport velocity can be estimated from the numerically simulated flow velocity based using the simple equilibrium equation.

For self-containment of this study, the derivation process of $u_b$ using the equilibrium equation can be summarized as:

The bed load transport velocity $u_b$ is determined by the tangential velocity component $u_{NB,T}$ near the bottom, and $u_{NB,T}$ can be defined by the outer product of the flow velocity vector near the bottom and the outward normal vector $N$ at the bottom as follows (see Figure 3):

$$u_{NB,T} = \frac{1}{|N|^2}[N \times (u_{nw} \times N)] \tag{27}$$

There are many external forces acting on the sediment particles lying on the bottom, and among these, gravitational force $w$, drag force $f_D$, and frictional force $f_f$ are dominating. Information concerned with $u_b$ can be obtained by assuming that the external forces mentioned above are balanced.

The gravitational force acting on the sediment can be written as:

$$w = \frac{\pi}{6}\rho(s-1)d^3 g \tag{28}$$

Via a similar approach to Equation (27), the tangential component $w_\tau$ of gravitational force along the bottom can be written as:

$$w_\tau = \frac{1}{|N|^2}[N \times (w \times N)] \tag{29}$$

where operator $\times$ denotes the cross product of vectors.

As the sediment particle is migrating with a velocity of $u_b$, the friction force acting in the opposite direction arises. Using dynamic friction coefficient $\mu_d$ and the vertical reaction force, the friction force can be written as follows:

$$f_f = -\frac{1}{|N|}|w \cdot N|\mu_d \frac{u_b}{|u_b|} \tag{30}$$

Here, the unit vector $-u_b/|u_b|$ reflects that the frictional force is acting in the opposite direction of $u_b$. Finally, the flow induced drag force $f_D$ can be written as:

$$f_D = \frac{1}{2}\rho c_s \frac{\pi}{4}d^2|u_r|u_r \tag{31}$$

where $u_r$ represents the relative flow velocity with respect to the moving sediment particle and is given by $u_{nw,\tau} - u_b$, $c_s$ is the coefficient that accounts for the weight loss due to the buoyancy of submerged sediment particles. Following Luque [23], $c_s$ can be written as:

$$c_s = \frac{4\mu_d}{3a^2 \frac{1}{2}\theta'_c} \tag{32}$$

From Equations (29)–(32), the bed grain velocity can be obtained using the equilibrium equation which can be defined as:

$$w_\tau + f_f + f_D = 0. \tag{33}$$

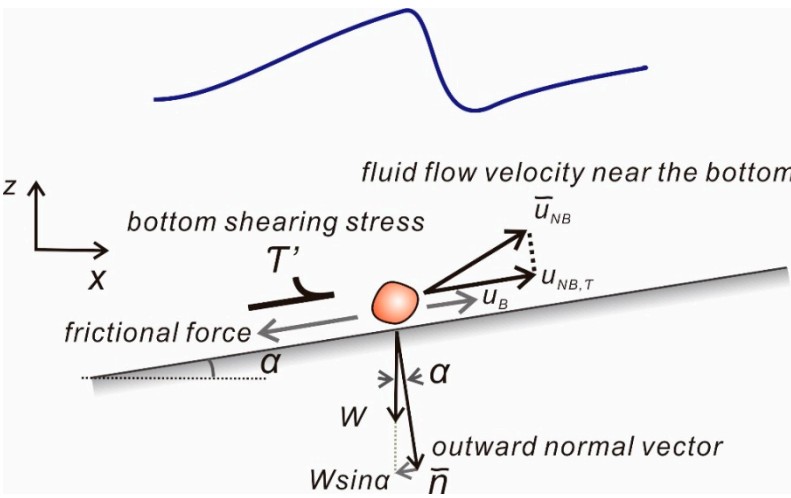

**Figure 3.** Free body diagram of the sand particles resting at the sea bottom of slope $\alpha$.

### 2.7. Morphology Model

If the net-flux of bed load along the seabed is not balanced with the erosion and deposition, there will be some change in bed morphology. The morphology model can be derived from the sediment budget principle applied to the control area which is depicted in Figure 4.

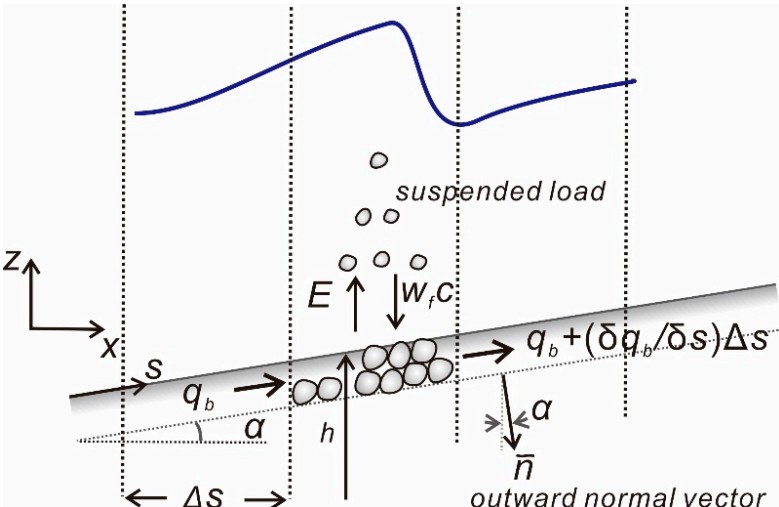

**Figure 4.** Schematic sketch of morphology changes of the seabed of slope $\alpha$ due to the erosion, and deposition process of suspended load, and the net flux of bed load.

The morphology model derived from the sediment budget principle can be written as:

$$\frac{\partial h}{\partial t} = -\frac{1}{1 - e_d}[\nabla \cdot q_b + E + D] \tag{34}$$

Equation (34) is of Exner type where $e_d$ is the porosity, $E$ is the amount of sedimentation that leaves the bottom by erosion, and $D$ is the amount of sedimentation that returns to the bottom due to gravity.

In Equation (34), following Fick's law, $E$ can be written as:

$$E = -(\nu + \nu_{tub})\frac{\partial c}{\partial n}|N| \tag{35}$$

where $\nu$ and $\nu_{turb}$ is the kinematic viscosity coefficient and eddy viscosity coefficient, respectively, and the negative sign is introduced to correct the difference between the directions of outward normal vector and erosion (see Figure 4). After denoting the unit outward normal vector at the bottom as $n$, the outward normal vector $N$ can be written as:

$$N = |N|n \tag{36}$$

where $|N|$ is the surface area of the hexahedron cell constituting the seabed. In Equation (34), $D$ can be estimated by projecting the summation of the falling velocity of the sediment particles and flow velocity $u$ on the outward normal vector $N$ at the bottom, and can be written as:

$$D = c_b\left|\left(w_f + u\right) \cdot N\right| \tag{37}$$

where $c_b$ is the sediment concentration at the bottom defined in Equation (25).

The change of the bottom height $\Delta h_b$ in the vertical direction during $\Delta t$ by $q_b$ is described as follows:

$$\Delta h_b = -\frac{1}{1 - e_d}\frac{\nabla \cdot q_b}{|n \cdot e_g|}\Delta t \tag{38}$$

where $n$ is the unit outward normal vector at the bottom in Equation (35), $e_g$ is the unit vector in the positive $z$ direction, and $\Delta t$ is the time step used in the numerical simulation. The net flux of the bed load along the inclined sea bottom during $\Delta t$ is given by $\nabla \cdot q_b \Delta t / 1 - e_d$, and hence, by dividing the net flux of the bed load by $|n \cdot e_g|$, the change of the bottom height $\Delta h_b$ along the vertical direction can be obtained.

## 3. Numerical Model

Numerical simulations are implemented using the IHFOAM toolbox having its roots on the OpenFoam [24]. In the IHFOAM, the wave model comprises the RANS and mass conservation equation, and the free surface is traced using the VOF (volume of a fraction) method. As the turbulence closure, the $k$-$\varepsilon$ model was used.

*Wave Model*

The RANS and mass conservation equation that constitute the wave model in IHFoam [24] can be written as:

$$\nabla \cdot U = 0 \tag{39}$$

$$\frac{\partial \rho U}{\partial t} + \nabla \cdot (\rho U U) - \nabla \cdot \left( \mu_{eff} \nabla U \right) = -\nabla p^* - g \cdot X \nabla \rho + \nabla U \cdot \nabla \mu_{eff} \tag{40}$$

where $U$ is the flow velocity vector, $g$ is the gravitational acceleration vector, $X$ is the position vector, $\mu_{eff} = \mu + \rho \nu_{turb}$, $\mu$ is the dynamic viscosity coefficient, $\nu_{turb}$ is the eddy viscosity coefficient, and $p^*$ is the quasi dynamic pressure.

The VOF equation for the analysis of two-phase flow can be written as:

$$\frac{\partial \gamma}{\partial t} + \nabla \cdot U \gamma + \nabla \cdot U_c \gamma (1 - \gamma) = 0 \tag{41}$$

where $\gamma$ is the VOF coefficient.

## 4. Numerical Results

In order to test the feasibility of the morphology model presented in this study as the platform for the optimal design of the beach nourishment project, the beach restoration process by the infra-gravity waves underlying the swells in a mild sea was numerically simulated. The numerical wave flume used for numerical simulation is shown in Figure 5, and the computational domain depicted in Figure 5 was discretized using 250,000 grids. In doing so, some efforts were made to ensure sufficient accuracy by deploying the first node from the sea bottom within the viscous sub-layer defined by $z^+ \approx 4$, where $z^+ = u^* z / \nu$ is the dimensionless distance from the seabed, and $u^*$ is the frictional velocity. Considering the geomorphological characteristics of the east coast of Korea like that from the shoreline to the site 60–80 m away from the shoreline, the foreshore of the steep slope is formed, and then the low tide terrace with a depth of 8 m extends offshore, the incident waves are generated based on the Cnoidal wave known as a representative nonlinear wave model at finite depth [1,25,26]. The depth was selected to be 0.8m to avoid excessive computational time and efficiently carry out the numerical simulation, which corresponds to the 1/10 scale. The wave conditions used for numerical simulation are listed in Table 1. The physical properties of the sediment particle were taken as $s = 2.4$ and $d = 0.02$ mm considering the characteristics of the sand constituting the beach along the east coast of South Korea [27].

The Cnoidal wave defined by the analytical solution of the Korteweg-de Vries equation can be written as:

$$\zeta = H \left[ \frac{1}{m} \left( 1 - \frac{E_M}{K_M} \right) - 1 + Cn^2 \left[ 2K_M \left( \frac{x}{L} - \frac{t}{T} \right) \middle| M \right] \right] \tag{42}$$

$$\frac{c^2}{gh} = 1 + \frac{H}{Mh}\left(2 - M - 3\frac{E_M}{K_M}\right) \tag{43}$$

$$\frac{HL^2}{h^3} = \frac{16}{3}MK_M^2 \tag{44}$$

$$c = \frac{L}{T} \tag{45}$$

Here, $\zeta$ is the sea surface displacement, $H$ is the wave height, $Cn$ is the Jacobi elliptic function, $c$ is the wave velocity, $L$ is the wavelength, $T$ is the wave period, and $M$ is the elliptic parameter ranging from 0 to 1. $K_M$ is the first type elliptic integral, $E_M$ is the second type elliptic integral, and $K_M$ and $E_M$ are dependent on value of $M$. It is known that as $M$ is converging to 0, Cnoidal wave reduces to Stokes 1st order wave, while $M$ is converging to 1, Cnoidal wave reduces to solitary wave.

**Table 1.** List of wave conditions used in the numerical simulations.

| Cases | Slope | Iribarren NO | H | T | h | Breaking Type |
|-------|-------|--------------|-----|------|------|---------------|
| RUN 1 | 1:6 | 1.38 | 0.1 m | 2.1 s | 0.8 m | Plunging |
| RUN 2 | 1:6 | 0.97 | 0.2 m | 2.1 s | 0.8 m | Plunging |
| RUN 3 | 1:6 | 0.79 | 0.3 m | 2.1 s | 0.8 m | Plunging |

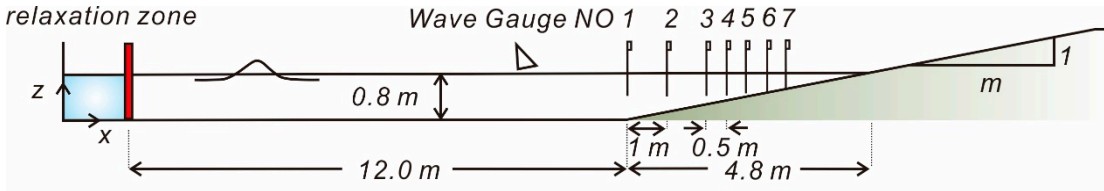

**Figure 5.** Computational domain and location of wave gauges [Gauge NO. 1: x = 12.0 m, Gauge NO. 2: x = 13 m, Gauge NO. 3: x = 14 m, Gauge NO. 4: x = 15 m].

Figure 6 shows the sampled time series of numerically simulated water surface displacement at varying stages of shoaling process. As can be expected, the asymmetric and skewed waves are observed due to the presence of higher-order harmonics generated during the nonlinear shoaling process. Figure 7 shows the temporal variation of the vertical profile of the shoreward velocity measured at Gauge NO. 1, 2, 3, and 4 that are equally spaced along the beach (see Figure 5). To integrate every facet of the cyclic flow over the beach slope, eleven frames per period are selected, and the averaged velocity profile over the unit wave period is also included for comparison. At Gauge NO. 1, and 2 where the shoaling process commences, the shoreward flow is more substantial than the offshore directed flow, and the enhanced Stokes drift can be found near the mean water level. On the other hand, at Gauge NO. 3 and 4, the offshore directed flow is much stronger than the shoreward flow and lasts longer than the shoreward flow, as can be expected in asymmetric wave profile with a steep front. It is also shown that the under-tow at Gauge NO. 3 and 4 is much faster than the one at Gauge NO. 1 and 2, and reaches its peak at Gauge No. 4, which is closest to a shoreline. Here, the other flow features worth mentioning are that at t = 90.6 s (Gauge NO. 3), the flow moves toward the offshore at the mid-depth. On the other hand, the flow near the free surface is directing toward the shore. It is also shown that at t = 91.2 s (Gauge NO. 4), the shoreward flow near the mid-depth moves more slowly than the one at the bottom boundary layer such that the shoreward flow reaches its minimum above the boundary layer. Above this retardation zone, the shoreward flow increases again, and the maximum shoreward flow takes place near the mean water level. These flow features, i.e., that the shoreward flow near the bottom moves faster than the one at the upper layer that is relatively free from bottom friction, are in line with the typical characteristics of boundary layer streaming [6,7], the presence of which was first identified by Longuet-Higgins [28].

For the case of period averaged flow, shore directed flow known as the Stokes drift is dominating near the free surface (see Figure 7a,b), and on the contrary, shore directed flow known as the under-tow is prevailing at mid-depth, implying that it is a return flow that determines the bed morphology over the surf and swash zone. The under-tow decreases as the water depth are deepened, but the spatial scope of under-tow extends toward the free surface. Here, it is worth recalling that if offshore directed flux along the beach is not balanced due to the weakened under-tow, these extra influxes are accompanied by the ascending flow that results in the formation of an offshore bar.

Figure 8 shows the temporal variation of the vertical profile of the shoreward velocity measured at the flat bottom (x = 6.0 m) in RUN 1 and RUN 2. As wave conditions are mild [RUN 1], boundary layer streaming is clearly visible near the sea bottom. On the contrary, as wave conditions are harsh (RUN 2), boundary layer streaming in the period averaged profile is noticeably weakened due to the more intensified under-tow.

Figure 9 shows the contour plot of the suspended load concentration within the water body after the beach being exposed to the waves for 100 s. As time goes by, the turbidity of the water increases, and in particular, the increase in the suspended load concentration over the surf and swash zone noticeably stands out.

When the infra-gravity waves underlying ever-present swells under mild sea conditions arrive at the wave breaking line, sediments dragged by offshore directed orbital wave motion enhanced by the gravitational force acting along the slope are rising near the water surface (see Figures 9c and 10), and then carried toward the top of the foreshore by the up-rush commencing from the front of breaking waves. During their passage across the surf zone, more material is suspended. Thus, uprush water is loaded with sand leading to the formation of swash bar.

Figure 11 shows the temporal variation of the vertical profile of the suspended sediment concentration. After 90 s, the water's turbidity reaches a steady-state, and suspended sediment concentration is exponentially decreased toward the mean water level.

Figure 12 shows the temporal variation of the bottom shearing stress at Gauge NO. 1, 2, 3, and 4. More considerable shearing stress is occurring during the run-up process than in the down-rush process. These features are frequently observed in the rapidly accelerating and slowly decelerating flow, such as the one under asymmetric waves with a steep front, and are known to be triggered by the premature boundary layer and its resulting thin thickness [22].

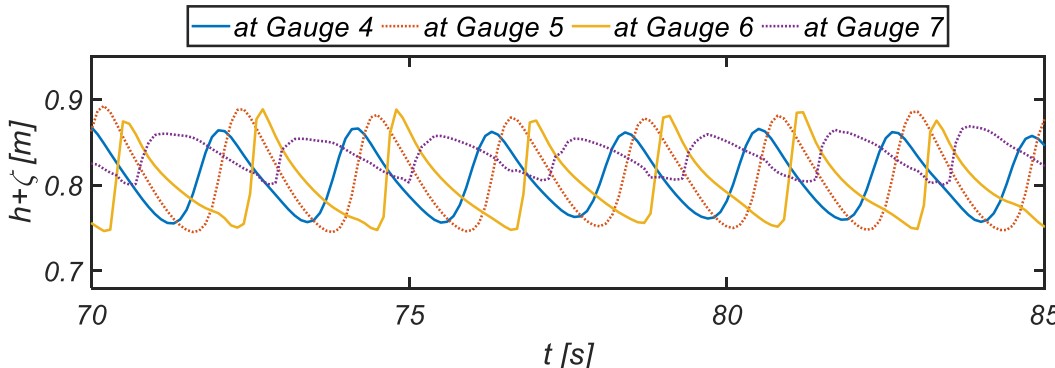

**Figure 6.** Sampled time series of numerically simulated free surface elevation at varying stages of shoaling process.

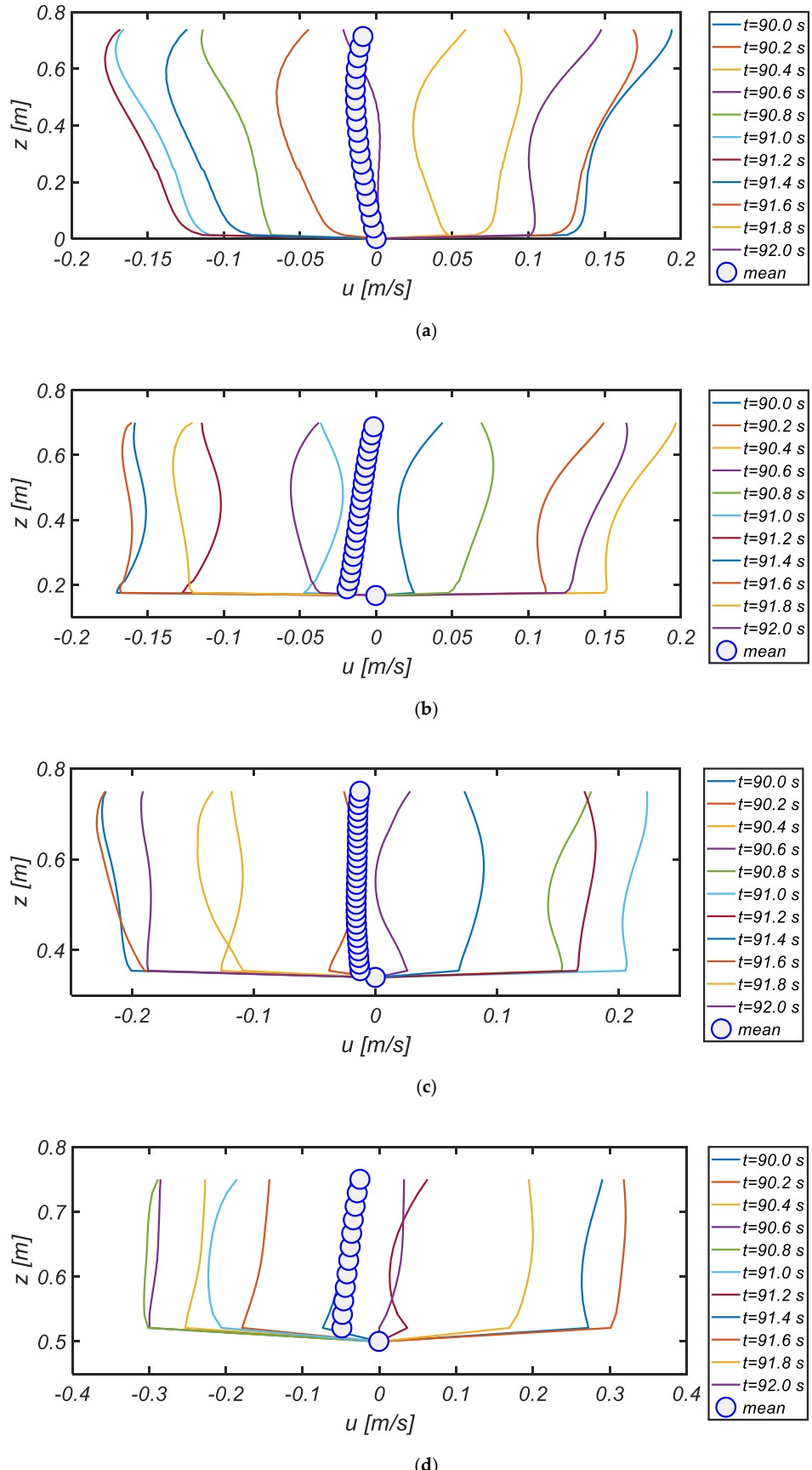

**Figure 7.** Temporal variation of the vertical profiles of horizontal velocity at Gauge No. 1, 2, 3, and 5 during unit wave period. (**a**) at Gauge NO. 1; (**b**) at Gauge NO. 2; (**c**) at Gauge NO. 3; (**d**) at Gauge NO. 5.

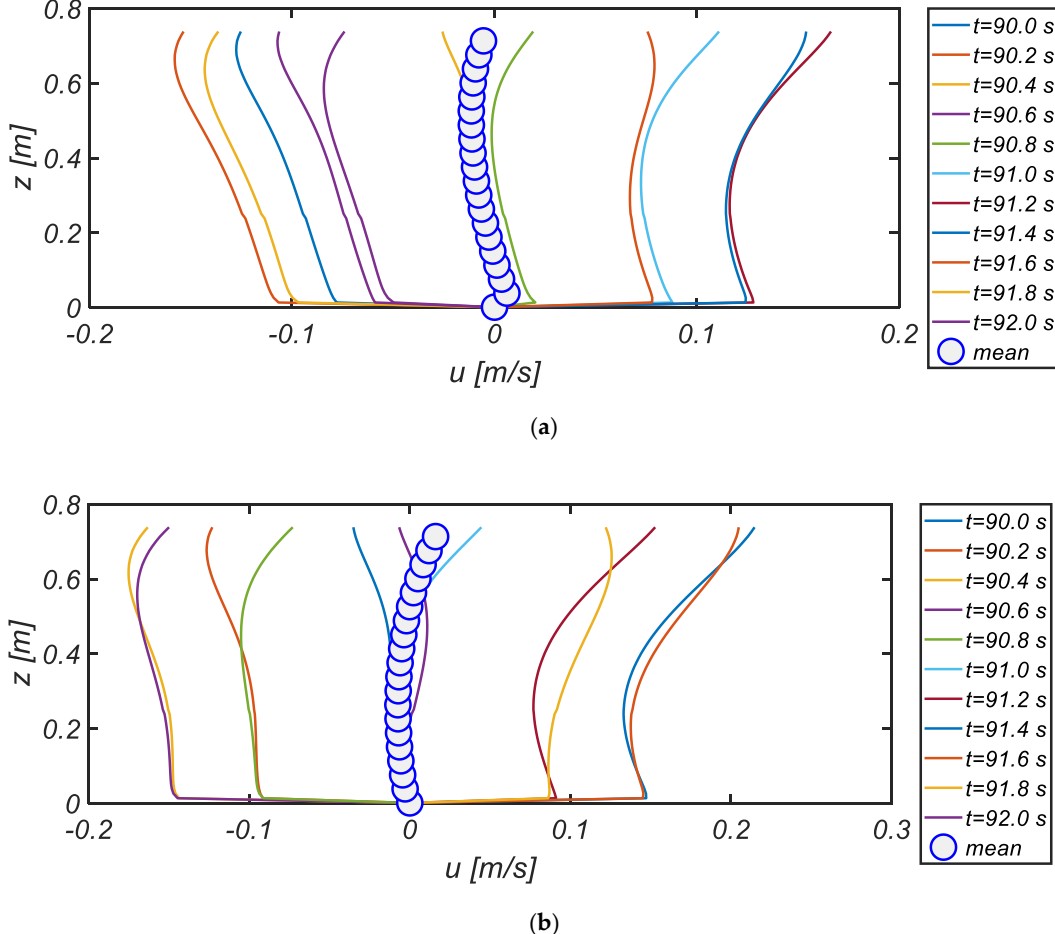

**Figure 8.** Comparison of the vertical profiles of horizontal velocity at x = 6.0 m during unit wave period by varying wave height. (**a**) RUN 1; (**b**) RUN 2.

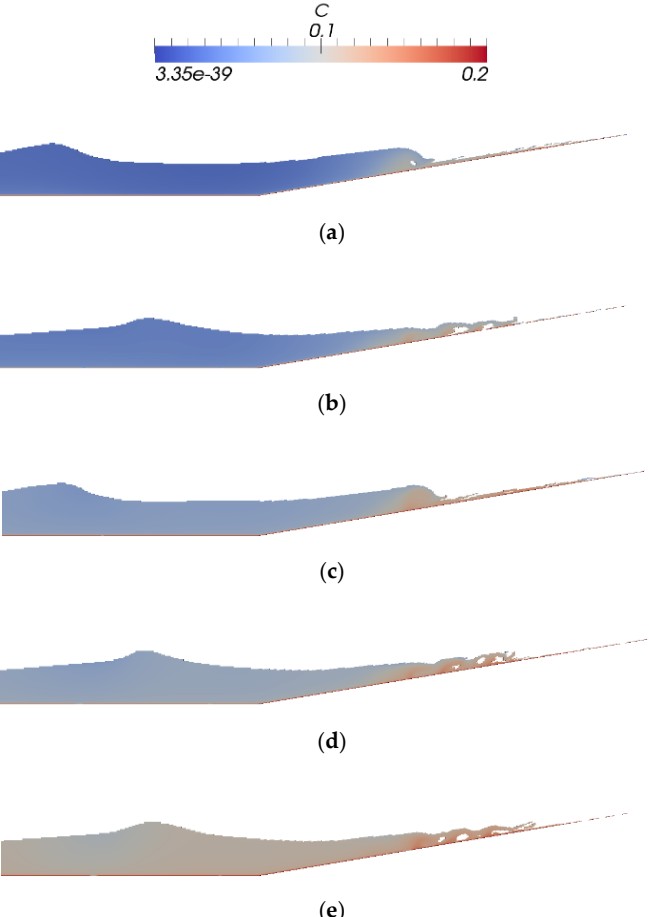

**Figure 9.** Sequential snapshots of numerically simulated suspended sediment concentration [Run 2]. (**a**) at t = 30 s; (**b**) at t = 40 s; (**c**) at t = 60 s; (**d**) at t = 70 s; (**e**) at t = 100 s.

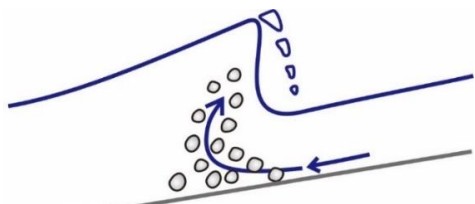

**Figure 10.** Schematic sketch of entrained sediment particles near the wave breaking line.

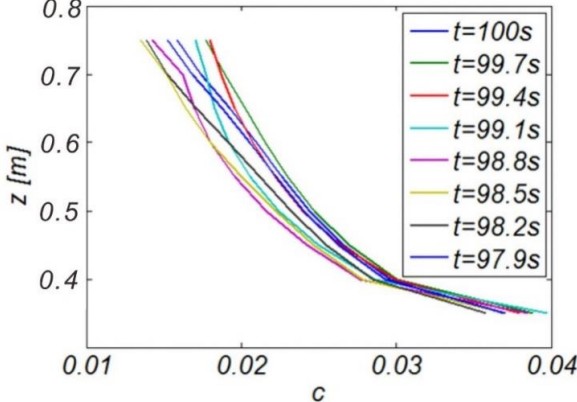

**Figure 11.** Temporal variation of vertical profile of the suspended sediment concentration [Run 2].

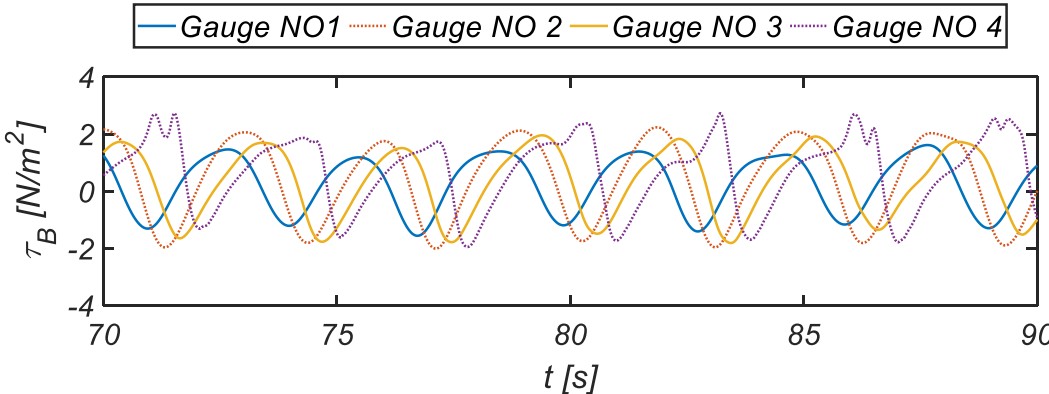

**Figure 12.** Temporal variation of the bottom shearing stress at Gauge NO. 1, 2, 3, and 4 [Run 2].

Figure 13 shows the bottom shearing stresses across the beach. The maximum shearing stress is occurring over the course of the run-up and reaches its maximum at $\tau_{Max} = 10\,\text{N/m}^2$. Considering that the conventional quadratic frictional law with frictional coefficient gives $\tau_{Max} = 1\sim5\,\text{N/m}^2$, it can be easily perceived that the morphology model based on the quadratic frictional law in the past studies would fall short in describing the flow over the surf and swash zone. The evaluation procedure of $\tau_{Max}$ based on the quadratic frictional law can be summarized as:

$$\tau_{Max} = \frac{1}{2}f\rho U_\infty{}^2 \tag{46}$$

where the free stream velocity $U_\infty$ and the friction coefficient $f$ can be written as:

$$U_\infty \approx \sqrt{gh}\frac{H}{h} \tag{47}$$

$$f = 0.04\left(\frac{A_b}{r}\right)^{-1/4} \tag{48}$$

In Equation (48), the semi-excursion amplitude $A_b$, and the Nikuradse roughness height $r$ are already defined in Equation (7).

Figures 14 and 15 show the bed morphology after the nourished beach exposed to waves for 100 s. In RUN 2 and RUN 3, where wave conditions are relatively harsh, sand waves formed over the flat bottom. Sand waves over the flat bottom are of sinusoidal shape under relatively mild wave conditions (RUN 2). On the other hand, sand waves became asymmetric as wave conditions are harsh (RUN 3). Under mild wave conditions (RUN 1), accretion is underway near the shoreline, and as a result, berm with steep slope appears near the shoreline, and these accretions were accompanied by eroded foreshore where wave breaking was taking place. Considering that numerical duplication of sand waves formed on the flat bottom would be the benchmark test of the physics-based 3D morphology model, the sand bar formed over the swash zone can be regarded to have a robust physical basis as well. Migration pattern of swash bar such that the size of swash bar is increased and moving toward the offshore as wave conditions is harsh is in line with our expectations based on the Deans number $\Omega$ telling that in the reflective beaches classified as $\Omega < 1$, the slope of foreshore gets steep and shoreline advances toward the offshore due to the sediments accreted over backshore in the process of a run-up in a mild sea.

As wave condition gets harsh [RUN2], the height of swash bar increases, and slightly migrating toward the offshore direction. On the contrary, the size of offshore bar diminishes, and the sand waves or ripples formed over the flat bottom are also migrating toward the offshore direction. However, the wavelength of sand waves is sustained to be the same in RUN 1 and RUN 2. These characteristics of sand bars also comply with the consensus reached in the coastal engineering community telling that

sediments entrained under the surface nodes of standing waves, where the near-bed velocities are most substantial, due to partially reflected waves from the beach slope are gradually drifted toward the surface antinodes of standing waves by the boundary layer streaming. As a result, the sand bars are formed at the antinodes of standing waves.

Summarizing the discussions mentioned above, the complexity of wave-induced flow reaches its extreme at the final stage of the shoaling process. The flow features like the Stokes drift near the free surface, offshore directed flow at the mid-depth, and shoreward boundary layer streaming near the bottom are far beyond the scope of the wave drivers based on the depth-averaged approach like the Boussinesq-type equation and NSW considering the assumption made in their derivation.

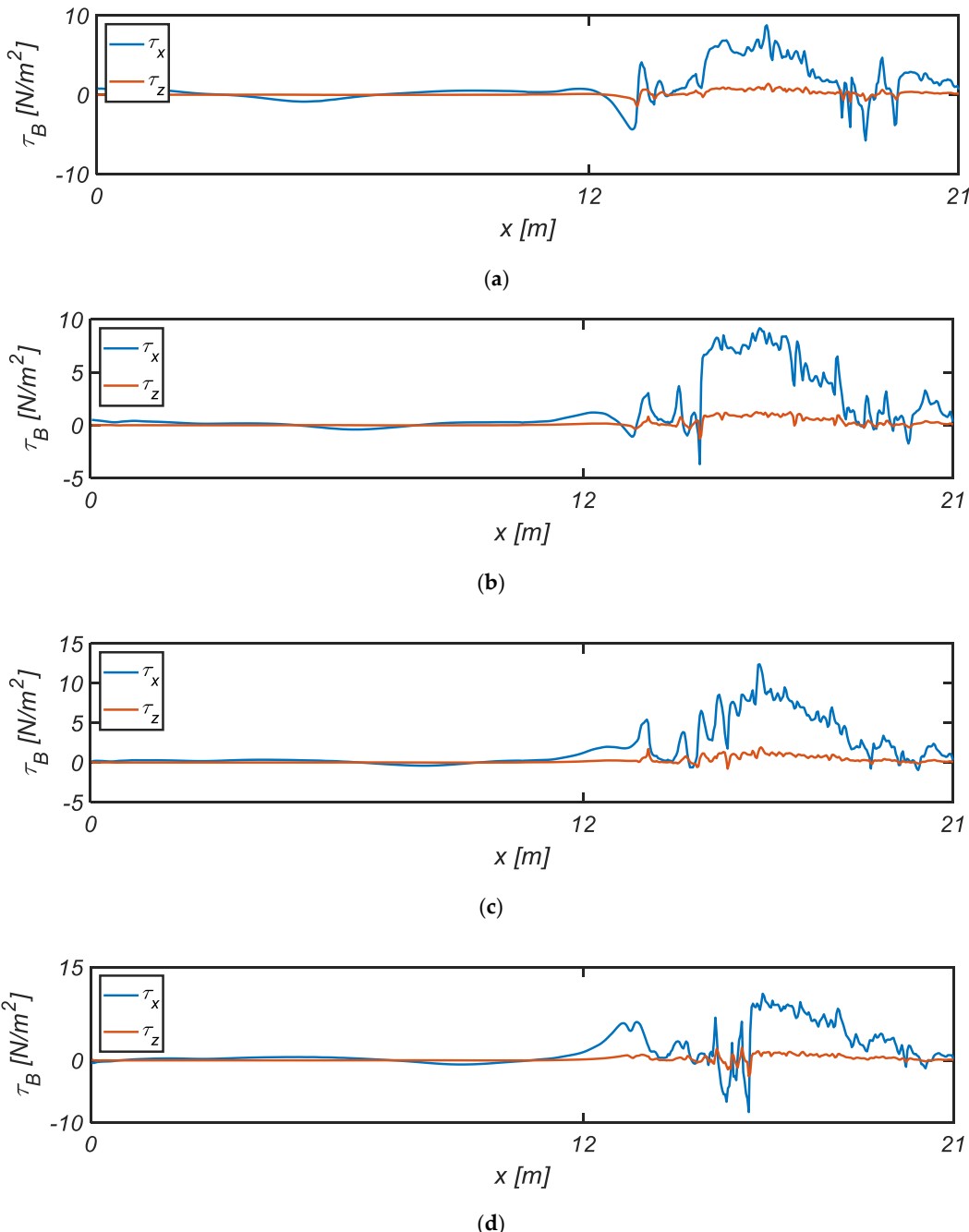

**Figure 13.** Variation of the bottom shearing stresses across the beach [Run 2]. (**a**) at t = 78.55 s; (**b**) at t = 79.15 s; (**c**) at t = 79.55 s; (**d**) at t = 80.0 s.

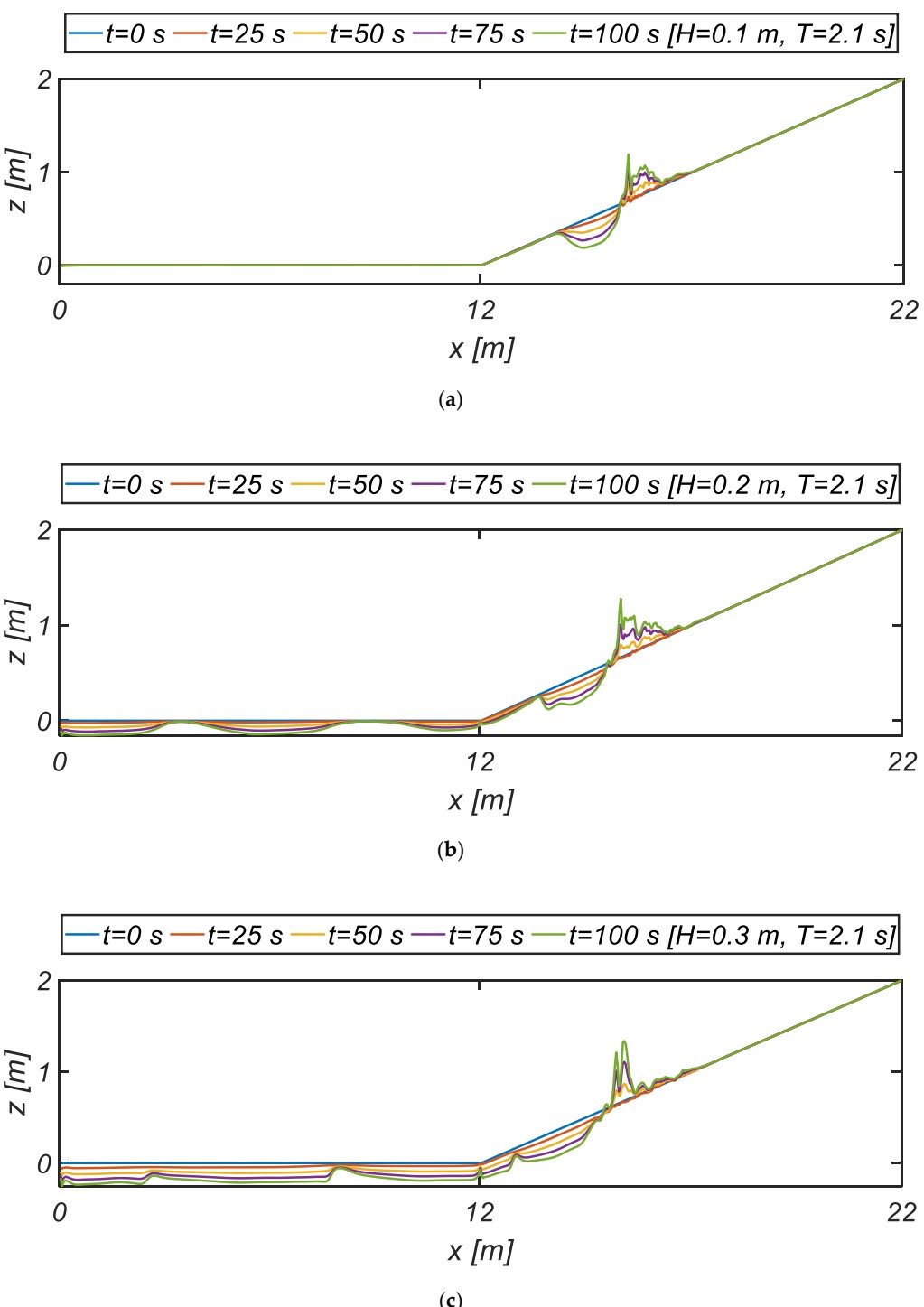

**Figure 14.** Evolution of beach profile as the number of attacking waves is accumulated. (**a**) RUN 1; (**b**) RUN 2; (**c**) RUN 3.

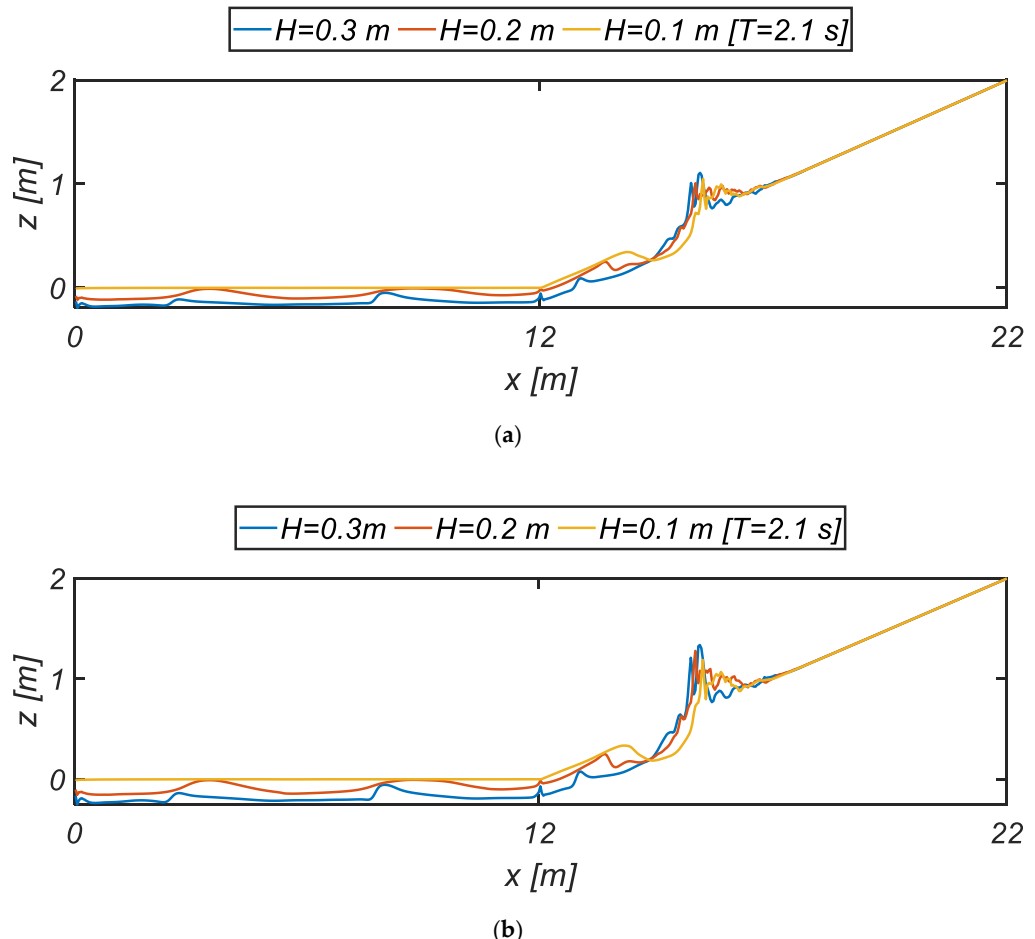

**Figure 15.** Comparison of beach morphology's change as the number of attacking waves is accumulated. (**a**) at t = 75 s; (**b**) at t = 100 s.

## 5. Conclusions

Natural beaches endlessly change their shape in response to changes in the sea conditions, and occasionally in response to human interference, such as the construction of breakwaters and groins. If the quasi-equilibrium water environment is preserved, the temporarily eroded beach by high waves can be restored by itself. Unfortunately, the quasi-equilibrium water environment along the east coast of South Korea has been damaged by poorly executed development, and as a result, beaches are severely eroded. Over the last decades, many efforts to cope with these erosion problems along the east coast of South Korea were made based on the traditional beach stabilization methods like a detached breakwater, LCB, groin, and jetty. Unfortunately, the beach stabilization effects of these structures as a countermeasure against beach erosion often fall short. Lately, the beach nourishment is emerging as a promising alternative. However, beach erosion cannot be prevented by nourishment, and periodic re-nourishment is inevitable. As a result, an accurate prediction of the nourished beach's erosion rate is requisite for the efficient implementation of beach nourishment to preserve a beach close to the natural one.

In this rationale, a physics-based bed morphology model was proposed in this study, and the beach restoration process by the infra-gravity waves underlying the swells in a mild sea was numerically simulated to test whether or not the morphology model proposed in this study work as the platform for the optimal design of beach nourishment project. As a hydrodynamic module, the IHFOAM wave toolbox having its roots in OpenFoam is used. Speaking of the morphology model, a transport equation for suspended load and Exner type equation derived from the sediment budget principle with the bed load being accounted for constitute the morphology model. In doing so, the probability

theory first introduced by Einstein [15] and the physical model test by Bagnold [16] were used as the constituent sub-model of the morphology model, and the incipient motion of sediment was determined based on the Shields diagram. The instantaneous bottom shearing stress required to estimate the bottom volumetric suspended sediment concentration and the bedload transport rate was directly evaluated from the numerically simulated flow field rather than the traditional quadratic Law and friction coefficient, which have been preferred in the past studies on the beach morphology.

Numerical results show that the partially skewed and asymmetric bottom shearing stresses were successfully simulated. Among many flow features that are indispensable in forming sand bars over the flat bottom and swash zone, a shoreward Stokes drift near the free surface, boundary layer streaming near the seabed, and undertow toward the offshore were successfully simulated using the morphology model proposed in this study. It also turns out that sand bars formed over the flat bottom in the numerical simulations, which have been regarded as a challenging task in the coastal engineering community, have a robust physical basis. The rationale for this reasoning can be found at the numerically simulated wavelength of sand waves that was sustained to be the same in the numerical simulations with varying wave height but identical wave period. On the other hand, the shape of sand bars gets more skewed toward the offshore direction, and the location of the sand bar is moving toward the offshore direction as wave conditions are harsh. These characteristics of sand bars are also in line with the consensus reached in the coastal engineering community that sediments entrained under the surface nodes of standing waves, where the near-bed velocities are most energetic, due to partially reflected waves from the beach slope are gradually drifted toward the surface antinodes of standing waves by the boundary layer streaming. As a result, the sand bars are formed at the antinodes of standing waves.

Considering that numerical duplication of sand waves formed on the flat bottom would be the benchmark test of the physics-based 3D morphology model, the sand bar formed over the swash zone can be regarded to have a robust physical basis as well. The migration pattern of swash bar, such that the size of the swash bar is increased and moving toward the offshore as wave conditions are harsh, also complies our expectations based on the Deans number $\Omega$ telling that in the reflective beaches classified as $\Omega < 1$, the slope of foreshore gets steep and shoreline advances toward the offshore due to the sediments accreted over backshore in the process of a run-up in a mild sea.

In the light of the facts mentioned above, it can be easily perceived that, with the phase resolving wave drivers like RANS (Reynolds averaged Navier–Stokes equation) coupled with the physics-based morphology model having the probability theory first introduced by Einstein [15] and the physical model test by Bagnold [16] as the constituent sub-model, sediments are suspended at the foreshore by wave breaking to be gradually drifted toward a shore and accumulate in the process of up-rush, which eventually leads to the formation of a swash bar, can be numerically reproduced. Considering that a swash bar is a distinctive feature of the temporal variation of a nourished beach over the course of a year, the morphology model presented in this study can provide the information like the erosion rate of a nourished beach, that is of great engineering value for the optimal design of the beach nourishment project. However, it is worth mentioning that the physics-based morphology model proposed in this study is partially verified against the moderate swells observed on the eastern coast of Korea and relatively fine sand. As a result, the robustness of physics-based morphology model is subject to further tests.

**Funding:** This research received funding as stated in Acknowledgments.

**Acknowledgments:** This research was a part of the project titled 'Practical Technologies for Coastal Erosion Control and Countermeasure', funded by the Ministry of Oceans and Fisheries, Korea. The author would like to thank the Korean Ministry of Oceans and Fisheries for funding this project.

**Conflicts of Interest:** The author declares no conflict of interest.

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
