# Peer review of "Development of the Physics–Based Morphology Model as the Platform for the Optimal Design of Beach Nourishment Project: A Numerical Study"

_jmse, doi:10.3390/jmse8100828_

Round 1

Reviewer 1 Report

This work proposes and applies a bed morphology model to reproduce the evolution of the beach profile under different wave conditions. From my point of view, the study is interesting and novel. However, major revisions are required to improve the preciseness, structure, readability and quality of the work. The main issues are detailed below:

- Why the subject ‘we’ is used in the text if the paper is signed by only one author?

- The English usage should be exhaustively revised throughout the manuscript. Working with a Scientific Editorial Company could be useful.

- Line 9. Replace ‘the physics-based morphology model’ by ‘a physics-based morphology model’.

- Line 11: ‘in this study work’. Remove ‘study’ or ‘work’.

- The last 3 keywords should be modified

- The introduction should be modified as follows: First, one or two paragraphs of general motivation of the work. Then, a review of previous works that deal with experimental measurements or numerical modelling of beach nourishment. Finally, the objective of the work and the structure of the manuscript.

- Lines 42-43. Add references to some works focused on analysis and/or modelling of beach evolution after nourishment as examples of previous progresses on the paper topic, such as:

Van den Berg, N., Falqués, A., & Ribas, F. (2011). Long-term evolution of nourished beaches under high angle wave conditions. Journal of Marine Systems, 88(1), 102-112.

Bergillos, R. J., Ortega-Sánchez, M., Masselink, G., & Losada, M. A. (2016). Morpho-sedimentary dynamics of a micro-tidal mixed sand and gravel beach, Playa Granada, southern Spain. Marine Geology, 379, 28-38.

Bergillos, R. J., Rodríguez-Delgado, C., & Ortega-Sánchez, M. (2017). Advances in management tools for modeling artificial nourishments in mixed beaches. Journal of Marine Systems, 172, 1-13.

de Schipper, M. A., de Vries, S., Ruessink, G., de Zeeuw, R. C., Rutten, J., van Gelder-Maas, C., & Stive, M. J. (2016). Initial spreading of a mega feeder nourishment: Observations of the Sand Engine pilot project. Coastal Engineering, 111, 23-38.

Bergillos, R. J., López-Ruiz, A., Principal-Gómez, D., & Ortega-Sánchez, M. (2018). An integrated methodology to forecast the efficiency of nourishment strategies in eroding deltas. Science of the Total Environment, 613, 1175-1184.

- The structure of the manuscript should be modified as follows:

1. Introduction

2. Methodology

        2.1 Wave model

        2.2 Sediment transport model

3. Results

4. Conclusions

- Section 2.3. The following work proposed a modification of Shields parameter to take into account the effect of bed slope and could be quoted in the section:

McCall, R. T., Masselink, G., Poate, T. G., Roelvink, J. A., & Almeida, L. P. (2015). Modelling the morphodynamics of gravel beaches during storms with XBeach-G. Coastal Engineering, 103, 52-66.

- The numbering of the figures is incorrect.

- Section 3.1. Much more details of the set-up and application of the wave model are required.

- I would include several subsections in the results section.

- Lines 358-383 should be moved to the methodology section.

- Line 399. A ‘[’ should be removed.

- Lines 525-537. This is introduction/motivation of the work, and not proper conclusions. Remove from the conclusions section.

- Lines 538-551. This is a summary of the methodology, so it could be also removed from the conclusions section.

- Add a final paragraph with the implications of the findings of this work for the international science and/or for management purposes.

Reviewer 2 Report

Review.

Overall, the manuscripts are well organized to describe the subjects, including detail theoretical backgrounds and numerical results. But I found several major comments to help clarify or improve the author’s conclusions in the manuscript.

Major Comment

  1. The scale of the wave period.
    The numerical model intends to simulate infra-gravity waves, as mentioned in Line 360. But the scaled wave period is 2.1 s, which is quite a short-wave period considering the scale of the numerical model (1/10).
  2. Need modifications in Fig. 7.
    Firstly, it is not a scaled sketch, so it is better to mention that it is an unscaled one. Secondly, Wave gauge No 1, 2, 3, 4 in Fig. 7 does not actually wave gauge (Wave gauge refer the measurement device to measure surface elevation in general). Fig. 8 shows the surface elevation at x=14.5 m to x=16.0 m, and this is not matched to the wave gauge described in Fig. 7. Readers will understand that wave gauges in Fig. 7 are for Fig. 8 (vertical velocity profile), but the author needs to clarify this in figure 7.
  3. In Fig. 8, the author concluded that the numerical model results show Stokes drift. One of the questions I had here, if there are a clear stokes drift, the mean u-velocity profiles will be positive overall. However, the result in Fig. 8 shows mostly negative mean values. Why it shows mostly negative u velocities?
  4. Mesh size and sensitivity.
    I need some clarification on the vertical mesh size. Especially at the bottom, the mesh size will be susceptible to the shearing stress in the numerical model. However, the mesh size is not given here. Different mesh sizes would be sensitive to the overall results in Fig. 9 to 17.
  5. In Fig 15. Why the author chooses these time steps to show Tau? (t=78.55 to 80.0s), and how does the author sure that this time presents the maximum Tau? Also, there are little descriptions for this figure. It would be better to add more descriptions. Otherwise, it would be better to delete this figure and addressed the maximum tau.
  6. Coupling wave and morphology model.
    Were fig. 17 results come from the 2 ways coupling model between wave and morphology model? If this is the coupled model, the author needs to detail how to couple them.
  7. Numerical model validation.
    Overall, the results in Fig. 17 show a reasonable sand migration pattern in the surf zone and near the coastline. Still, there is no quantitative or qualitative validation process for this numerical model setup. Otherwise, please describe this limitation of the current approach in the discussion.

Minor

  1. Typo in EQ 24, θc
  2. At fig 3 and 4. The sketch of the wave profile seems unnecessary. It is also repeated.
  3. What is the model run case of Fig. 11, 13, and 15?
  4. 15 needs information on X-axis (label needs), and please restate line 470 with more detail datum value (Tau_max is 10.xxx at t=7x.xx sec).

Reviewer 3 Report

The paper entitled Development of the physics-based morphology model as the platform for the optimal design of beach nourishment project: a numerical study reports on the feasibility of the morphology model presented as the platform for the optimal design of a beach nourishment project – using numeric simulation.

The experiments seem to have been well conceived and conducted, and are particularly interesting, as they try to overcome the problems with periodic nourishment activities, by trying to predict how the beach is going to erode.

The paper is also well organized and very well written – including an extensive and very complete explanation of different models in the introduction. Nevertheless, a small number of points need to be clarified:

  • While it states the article only has one author (Yong Jun Cho), it sometimes is stated “we” or “our” (e.g.: line 9 and line 201, respectively). I would advise writing in an impersonal tense, without mentioning singular or plural subjects.
  • Both in the keywords and the references, a different font was used. The font should always be Palatino Linotype.
  • Some paragraphs in the literature review chapters are missing references (e.g.: lines 39-41; line 49; lines 54-62). Even if it is considered “common knowledge”, it is always very important to include references.
  • While the final chapter of the introduction explains the main goals of the study, it would be interesting to also include a small description of each chapter of the article (i.e.: explain the structure of the article).
  • While it is stated the study is based on the east coast of Korea – I wonder if that is not too vague of a location? Or if the location should be specified in a small chapter as well.
  • The conclusion is also very complete and makes a through analysis of all the topics studied on the article. However, it should focus more on “future developments”, and maybe summarize the first few chapters – which are almost a word-by-word repetition of what has already been mentioned in the introduction.

The attached pdf file includes all this information in the respective areas, also including a few more small points and suggestions. I suggest the author revised carefully this small list of points and, after that, the paper can be published without any further review.

Round 2

Reviewer 1 Report

The author has not addressed most of my comments. Extensive editing of English language and style are still required, as I suggested in the previous revision. The present version of the manuscript has not enough quality to be published in JMSE.

Thus, I regret to inform you that my recommendation is to reject this work.

Author Response

thank you for your valuable comments

Reviewer 2 Report

Thanks for the quick updates. 

The author clarifies most of the questions I have and I feel it is right to publish in JMSE.

Author Response

thank you for your valuable comments